# Regulation Intensity, Freedom of Production Decision and the Poverty of Farmers: Evidence from the Panda Nature Reserves in China

**Chao Lin and Lan Gao ***

Department of Forestry Economic and Management, Economics & Management College, South China Agricultural University, Tianhe District, Guangzhou 510642, China; lin2021chao@gmail.com
* Correspondence: gaolan@scau.edu.cn

**Abstract:** The harmonious coexistence between human and nature is a topic of general concern. Existing studies generally agree that the establishment of nature reserves can effectively protect biodiversity, but less attention has been given to the effects of the regulation implied therein upon the multidimensional poverty of surrounding farmers. This paper verified the effects of regulation intensity upon the multidimensional poverty of farmers at the two levels of theory and empirical evidence from the perspective of freedom of production decisions based on the research data involving farmers around the Panda Nature Reserves (PNRs) in China. We have found that regulation intensity will significantly increase the multidimensional poverty of farmers. Heterogeneity analysis indicates that the regulation of agricultural production and the regulation of pollutants will produce a significant positive effect on the multidimensional poverty of farmers. The inherent mechanism is that regulation affects farmers' freedom of production decisions, thereby worsening the state of their multidimensional poverty. The conclusion of this paper not only contributes to expanding the theoretical studies on regulations and the multidimensional poverty of farmers but also offers suggestions on how the Chinese government can strike a balance between ecological protection and the prosperity of farmers.

**Keywords:** regulation; poverty; production decision; panda nature reserves

## 1. Introduction

The elimination of poverty is a focus of the whole world. China has made achievements in the elimination of absolute poverty, which have attracted global attention and have been highly recognized by international organizations [1], such as the World Bank. According to the calculation based on the existing Chinese standard for rural poverty, the poverty count of China was 770.39 million in 1978 and the poverty incidence was 97.5%. By the end of 2019, the poverty count was reduced to 5.51 million and the poverty incidence was merely 0.6%. Furthermore, China has comprehensively built a moderately prosperous society, historically solving the problem of absolute poverty.

However, the elimination of absolute poverty does not mean a thorough solution to poverty. Generally speaking, poverty can be classified into three kinds: absolute poverty, relative poverty, and social exclusion. Under absolute poverty, individuals lack the resources that allow them to maintain basic survival. Relative poverty means individuals lack or have difficulty obtaining the resources necessary for their daily life. Meanwhile, social exclusion underlines the integration of individuals with society as a whole [2]. In terms of practice, China has now eliminated absolute poverty. However, relative poverty and social exclusion have not yet been solved, which means poverty reduction has entered a new stage in China.

To set up a long-term mechanism of solving poverty, the cause must be clarified. Different views on the cause of poverty can be found in academic circles. Some scholars

believe that objective and explicit factors are the key [1,3–6], e.g., system, policy, natural environment, geographical location and physical capital, human capital, and technology. These views are conscious of the effects of the external environment and system but have difficulty explaining why the income gap and relative poverty exist under the same restraint, which is why other perspectives have arisen, including poverty culture theory, social exclusion, and rights deprivation theory [7]. With the deepening of poverty studies and poverty reduction practices, the connotation of poverty has expanded from income poverty in a narrow sense to human poverty in a broad sense [8]. More attention has also been given to absolute income deficiency, and the political and academic circles have focused more attention on relative poverty and rights deprivation.

Sen (2000) [9] is the founder of the multidimensional poverty theory. Based on his theory, the United Nations Development Program (UNDP) [8] developed the human poverty index, which is used for the assessment of the poverty state. In determining solutions to poverty, the first step is to identify accurately who can be categorized as belonging to the poor. Existing studies have discussed how to measure poverty accurately. Compared with earlier studies, the current consensus in academic circles is that relative poverty will exist in the economic society for a long time and its measurement should be multidimensional, and thus, the unidimensional poverty standard theory has been abandoned. For example, Huo et al. (2021) [10] built an index system of physical, cultural, and social needs from the perspective of the needs of a good life for measurement of poverty. Some scholars believed that under the state of relative poverty, it is even more necessary to focus on the "poverty" in social development and, therefore, to set a composite measurement standard that reflects equal stress on the "poverty" in both development and the economy, as well as an appropriate number of rural people in relative poverty [1]. In addition, some scholars believe that with the development of the economic society, the indices of multidimensional poverty should also change, and therefore, poverty indices should be established dynamically to suit local conditions [4]. Existing studies generally focus on the state of relative and multidimensional poverty, highlighting the focus on the social integration and development of the poor.

However, the ultimate development of a society and individuals depends on realizing the harmonious coexistence between humans and nature [11]. Therefore, the governments of various countries generally reach the goal by setting up nature reserves. Since the first nature reserve was set up in 1956, China has formed a system of nature reserves with complete types, reasonable layout, and sound functions. By the end of 2018, China (excluding Hong Kong, Macao, and Taiwan) had set up 2750 nature reserves, including 474 at the national level, which have a total area of 1.47 million km$^2$, accounting for about 14.84% of the national land area [12]. Despite the recognized importance of the nature reserves in protecting biodiversity, studies have found a high degree of overlap between the nature reserves and the regions inhabited by national minorities and the poor [13,14]. Most farmers living in or around the nature reserves fall into the poor of lower living standards [15]. Therefore, scholars began to focus on the externalities of nature reserves for the surrounding farmers. The studies of some scholars show that the establishment of nature reserves will increase the poverty of the farmers living therein or nearby [13,16,17]. The reason is that nature reserves restrict the resources available to farmers in the process of protecting biodiversity [13], but policies have ignored the requirements of the surrounding farmers [18]. For example, with the vigorously boosted ecological projects, including the "Grain for Green Program", the protection of natural forests and the protection zone works caused serious loss of farmland to the surrounding farmers, thereby increasing the degree of poverty. Some scholars believe the establishment of nature reserves will affect the surrounding farmers [19–21] because they received earnings even though they bear the cost of environmental protection [4]. Direct earnings include the ecological compensation that the farmers can receive, the employment they find in the reserves, income from tourism, and reasonable collection of resources [22–24]. Additionally, the surrounding farmers can also receive indirect earnings, e.g., those derived from improvement from community

environment and infrastructure [25]. Some scholars even made direct verification to show that the establishment of nature reserves can help mitigate the poverty of surrounding farmers [26,27] because, on the one hand, economic development can drive farmers to lower their reliance upon natural resources and, on the other hand, a series of development projects carried out around the nature reserves can attract the preferential policies of the government [28].

In summary, although existing studies have focused on the relationship between the establishment of nature reserves and the poverty of surrounding farmers, most studies have conducted their analysis only from a dichotomic perspective, i.e., "whether" the farmers are in a reserve and "whether" they are under control and neglected the regulation spillover from the establishment of nature reserves toward surrounding farmers. Therefore, they seldom distinguish between the effects of different regulation intensities on the poverty of farmers. Moreover, early studies are insufficient in their understanding of poverty and more interested in absolute income poverty or livelihood, with less care as to the effects of the multidimensional poverty of farmers. Compared with the unidimensional income poverty, only multidimensional poverty, which is measured from multiple dimensions, can dissect the essence of poverty [29]. On this account, this paper uses the farmer survey data of Panda Nature Reserves (PNRs) in China to dissect the effects of its regulation intensity upon the multidimensional poverty of surrounding farmers and its internal mechanism from the two levels of theory and empirical evidence. The paper not only expands the theoretical research on regulation and multidimensional poverty but also makes policy suggestions on promoting the ecological civilization of China and achieving common prosperity.

## 2. Materials and Methods

### 2.1. Theoretical Analysis

In developing countries or underdeveloped regions, farmers constitute the most basic and major economic organization. Generally speaking, farmers live in rural areas and make production decisions by using limited capital goods under an established technical level. Property right theory indicates that resources are the wellhead whereby individuals receive earnings and the earnings from resources are sourced from the rights attached thereto. The value of rights decides the value of resources [30] and the right for individuals to own resources decides the earnings they receive from the resources. Therefore, when the rights attached to resources are restricted, especially when the use of crucial resources is restricted, the scale of labor division and the earnings of individuals will decrease. The essence is that the regulation of the rights to resources makes it difficult for the owners of property rights to make reasonable production decisions, thereby reducing the supply of labor and the allocation of resources causing the owners to ultimately fall into the vicious spiral of poverty.

For the farmers around a nature reserve, most of their earnings are sourced from the environment of the nature reserve, including income from the direct use of environmental products, income from the activities based on natural resources, and income from ecological compensation [31–33]. Most of all, in developing countries, the production decisions of the farmers around nature reserves rely on the consumption of environmental resources [34], such as the cash earnings obtained from the collection of wild food for replenishment of energy or collection of wood and medicinal materials [35]. However, in many countries or regions, especially in developing countries, nature reserves were established at the expense of the interests of the surrounding rural areas. Through the top-down legal restrictions, the right of farmers to receive earnings from the environment in a reserve is under regulation [36] and they are forced to change their mode of production [20], thereby causing serious negative effects, contradictions, and conflicts [37]. These negative effects are because temporary and permanent poverty may take place once the use of natural resources is restricted [38]. On the one hand, regulation has directly deprived surrounding farmers of their production rights and affected their freedom of production decisions, causing them to reduce the supply of agricultural production. On the other hand,

regulation has damaged the inherent incentive mechanism of surrounding farmers so that they lack the drive to leverage resources to take part in the division of labor and realize the optimized allocation of resources. When production opportunities are regulated, farmers' freedom of production decisions is restricted, which brings them towards the vicious spiral of poverty.

To alleviate the poverty of farmers, the government has taken a series of interventive measures, e.g., skill training and monetary support [39], and formulated matching support measures for employment, education, and medical care [40]. However, such "blood transfusion" assistance can only temporarily reduce the absolute poverty of farmers without reaching a fundamental solution to their relative poverty and social exclusion. If the rights of farmers are always under regulation, it will be impossible to realize the freedom of production decisions and difficult to arouse their internal impetus and achieve all-round elimination of poverty. Thus, this paper probes into the effect of regulation intensity upon the multidimensional poverty of surrounding farmers from the perspective of freedom of production decisions in the hope of offering policy suggestions to assist the country in solving relative poverty, effectively protecting biodiversity and achieving harmonious coexistence.

### 2.2. Source of Data

We applied the structural questionnaire survey of farmers (face-to-face interview method) to survey the 17 local communities around the 5 PNRs in Shaanxi (Figure 1), and 48 local communities around the 12 PNRs in Sichuan in July and October 2018 and January and May 2019, respectively. The communities in a nature reserve were sequenced based on their economic development level and per capita annual income and divided into two groups to randomly select one community from each group. The communities outside the nature reserves were processed in the same way. Four communities (two inside and two outside) in one PNR in total. However, because some nature reserves do not have 4 communities, 60 communities were finally selected. Finally, 15 households were selected randomly from each community from the roster of all the village personnel provided by the local village collective. The questionnaires collected the data on two levels: villages and families. The village questionnaire collected information on the village's natural environment, geographical features, human environment, and economic development. From the questionnaire, we obtained information about family members, resource endowment, production and management, protection area settings, multidimensional poverty topics, etc. After the samples with missing key information were weeded out, 864 valid samples were obtained.

### 2.3. Definition of Variables and Descriptive Statistics

2.3.1. Dependent Variable: Multidimensional Poverty

Farmers' poverty is the integration of multiple dimensions, including society, politics, culture, and system [41]. Therefore, in addition to income, the indices of multidimensional poverty should also include objective indices, e.g., education, medical care, and drinking water, and the index of the personal subjective feelings of farmers [42]. First, in the creation of the indices of multidimensional poverty, the selection of dimensions is crucial, meaning that it is necessary to select the appropriate dimensions based on the area under survey to measure the conditions of local poverty. This paper consulted the multidimensional poverty indices in the UN Human Development Report and set up a system of indices for the multidimensional poverty of farmers by leveraging the A-F measurement method, which is the most widely used and the most complete method for measuring multidimensional poverty and drawing upon the practices of Alkire and Fang(2019) [43] and Duan et al. (2020) [44]. See Table 1 for the specific indices.

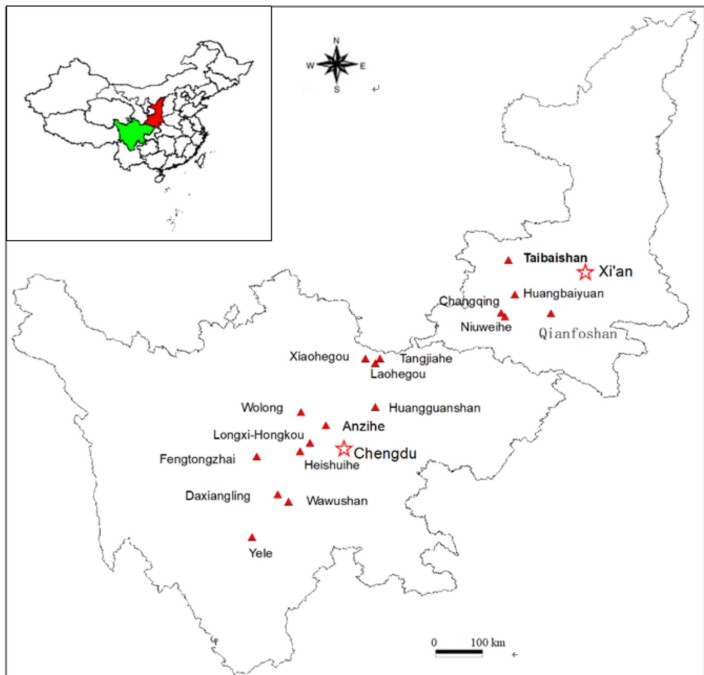

**Figure 1.** The distribution of PNRs in Shaanxi and Sichuan province.

At present, the index weight assignment methods of multidimensional relative poverty are divided into the non-equal weight method and equal weight method. Among them, the non-equal weight method includes the principal component analysis method and entropy method. The equal weight method includes dimension equal weight and index equal weight. Among them, the principal component analysis method requires a greater correlation between the indicators, while the entropy method requires a greater degree of dispersion in the values of the indicators. The data analysis of this study found that the Pearson correlation coefficient between the 9 indicators in the dimensions of education, health, and living standards is less than 0.3, and the data correlation is low, and the values of the indicators are all 0 or 1, which is not suitable for the non-equal weight method.

Therefore, this study uses a simpler equal weight method to assign the weights of the multidimensional relative poverty index of farmers in the reserve. According to the equal-dimensional weighting method, the four dimensions of income, health, education, and living standards are assigned an average value of 0.25, respectively, and then a single indicator is assigned an average value according to the number of indicators in each dimension. According to the equal-index weighting method, 10 indicators are directly assigned an average weight, each of which is 0.1. The specific weighting situation is shown in Table 1.

From Table 1, in the system of measuring the multidimensional poverty of farmers, this paper selects four dimensions of income, health, education, and living standards to measure multidimensional poverty. Specifically, the income is measured by whether the family income is less than 2995 yuan; the health is measured by whether a family with anyone who is disabled or in serious illness has any inability to receive medical treatment in a regular medical institution after falling ill; the education is measured by whether the labor force has an average education of < 6 years and any uneducated family member age 6–16; the living standard includes drinking water, fuel, lavatory, housing materials, and disposal of garbage.

**Table 1.** System of measuring the multidimensional poverty of farmers.

| Dimension [1] | Index | Critical Value | Weight of Dimensions | Weight of Indices |
|---|---|---|---|---|
| Income | In poverty or not | For a family with a per capita annual income of <2995 yuan [2], the assignment is 1; otherwise 0 | 0.25 | 0.1 |
| Health | Incidence of serious illness and disability | For a family with anyone who is disabled or in serious illness, the assignment is 1; otherwise 0 | 0.125 | 0.1 |
| | Timely medical treatment | For any inability to receive medical treatment in a regular medical institution after falling ill, the assignment is 1; otherwise 0 | 0.125 | 0.1 |
| Education | Years of education | For the labor force with an average education of <6 years, the assignment is 1; otherwise 0 | 0.125 | 0.1 |
| | School attendance | For any uneducated family member aged 6–16, the assignment is 1; otherwise 0 | 0.125 | 0.1 |
| Living standard | Drinking water | For use of other water than tap water/barreled water/purified water/filtered water in cooking, the assignment is 1; otherwise 0 | 0.05 | 0.1 |
| | Fuel | For the exclusive use of firewood and straw in heating and cooking, the assignment is 1; otherwise 0 | 0.05 | 0.1 |
| | Lavatory | For a lavatory without flush, the assignment is 1; otherwise 0 | 0.05 | 0.1 |
| | Housing materials | For housing with an earth–timber structure, the assignment is 1; otherwise 0 | 0.05 | 0.1 |
| | Disposal of garbage | For the absence of fixed/classified garbage collection near the home, the assignment is 1; otherwise 0 | 0.05 | 0.1 |

[1] UNDP uses 10 indices in the three aspects of "health, education, and living standards" for weighted calculation of multidimensional poverty indices. [2] Because of the changes in commodity prices, year-based adjustments need to be made to the standard for poverty in different years. The 2018 standard was 2995 yuan. See "Poverty Monitoring Report of Rural China" for rural poverty standard.

Second, multidimensional poverty is identified to judge whether the deprivation of benefits that meets k number of dimensions (indices) at the same time exists. $k$ is the dimension. The selection of dimension $k$ is the key to the calculation of the indices for multidimensional poverty. Here, according to the aforementioned method of measuring multidimensional poverty, when the weight of the indices is equal and the selected value of $k$ is 1, 2, 3, 4, and 5, the incidence of poverty (abbreviated as H), degree of poverty deprivation (abbreviated as A), and indices of multidimensional poverty (abbreviated as M) are calculated to estimate the multidimensional poverty result of farmers with different ways of living. The different selected values of $k$ will lead to the different contribution rates of corresponding M and the 10 indices of four dimensions. For example, when the selected value of $k$ is 3, a farmer meeting any 3 of the 10 indices of poverty is in poverty. See Table 2 for the index measurement results of the multidimensional poverty of farmers.

Table 2 shows that different $k$-value multidimensional poverty indices are different, and no matter if from the dimension equal weight or the index equal weight, the $k$ value and the multidimensional poverty all show a positive correlation. In other words, the larger the $k$ value, the lower the multidimensional poverty index of the farmer households, which shows that when measuring the multidimensional poverty of the farmer households, the $k$ value needs to be selected based on the actual situation and should not be too small or too large. Therefore, based on the practices of Alkire and Fang (2019) [43] and the actual conditions, $k = 3$ is selected for analysis in this paper.

**Table 2.** Index measurement results of multidimensional poverty of farmers.

|  | Dimensional Equal Weights | Index Equal Weights |
|---|---|---|
| $k = 1$ | 0.078 | 0.069 |
| $k = 2$ | 0.042 | 0.032 |
| $k = 3$ | 0.017 | 0.013 |
| $k = 4$ | 0.006 | 0.004 |
| $k = 5$ | 0.001 | 0.001 |
| $k = 6$ | 0.000 | 0.000 |

2.3.2. Main Independent Variable: Regulation Intensity

Existing studies mostly measure regulation from a dichotomic perspective of "yes and no" and, by default, regard farmers outside the nature reserves as being free of control. However, the regulation policy has a spillover effect on the farmers outside the nature reserves [44], such that surrounding farmers will also be subject to some restrictions. Hence, this paper uses the entropy method to calculate the regulation intensity based on the differences in the changes of the regulation intensity that the farmers are exposed to. The regulation includes the regulation of agricultural production and the regulation of pollutants [45]. Specifically, the former includes regulations on the application of pesticides and chemical fertilizers, woodcutting, gathering of firewood, collection of wild medicinal materials, and grazing, while the latter includes the regulation of the emission of garbage and wastewater and the disposal of the excrement of humans and livestock. The specific assignments are 0 = nonexistent; 1 = lax; 2 = strict; and 3 = prohibited completely. It should be noted that regulation is a policy promoted by the government and usually implemented at the level of villages and is exogenic for individual farmers, which is why regulation intensity is deemed as an exogenous variable [46]. See Table 3 for the indices and weight.

**Table 3.** System of measuring the regulation intensity of nature reserves.

| Dimension | Index | Weight of Entropy Method |
|---|---|---|
| Regulation of agricultural production | Application of pesticides and chemical fertilizers | 0.343 |
|  | Wood cutting | 0.069 |
|  | Gathering of firewood | 0.086 |
|  | Collection of wild medicinal materials | 0.150 |
|  | Grazing | 0.195 |
| Regulation of pollutants | Emission of garbage and wastewater | 0.074 |
|  | Disposal of the excrement of humans and livestock | 0.082 |

It can be seen from Table 3 that after weighting by the entropy method, the weight of application of pesticides and chemical fertilizers index is the largest, followed by grazing, and the third is the collection of wild medicinal materials. The weights of the other indicators are all less than 0.1, and the weight of the wood cutting is the smallest.

2.3.3. Control Variables

Drawing upon existing studies, this paper uses the regulation variables, which include householder, family, and village characteristics. Specifically, householder characteristics include the age, gender, and years of education of a householder and whether the householder is a village cadre, while family characteristics include the number of family laborers, burdens of upbringing, proportion of non-farm employment, area of farmland, area of woodland, and the distance to the town market. In addition, controlling for village fixed effects in our empirical model also mitigates omitted variable bias caused by unobservable village characteristics.

2.3.4. Descriptive Statistics

1.  Incidence of the poverty under multidimensional indices for farmers in different geographical locations.

The established multidimensional poverty index system is used to compare the incidence of the multidimensional poverty of the farmers in different geographical locations (see Table 4). Drawing upon existing studies, this paper divides geographical location into the buffer zone, experimental zone, and outside area. Table 4 shows that on the whole, farmers in the buffer zone have the highest incidence of poverty, which can be reflected by the indices of income, the incidence of serious illness and disability, timely medical treatment, fuel, lavatory, and housing materials as compared with the relatively low incidence of poverty for farmers in the experimental zone. Following the legal provisions on nature reserves, only the activities of scientific research and observation are allowed in a buffer zone. More activities are allowed in an experimental zone, such as scientific experiments, teaching practices, visits, observations, tourism, and taming and breeding of rare wild animals and plants. It also indicates that the buffer zone is subject to more controls of higher intensity, which results in a higher incidence of farmers' poverty.

**Table 4.** Incidence of poverty under multidimensional indices for farmers in different geographical locations.

| Dimension | Index | Buffer Zone | Experimental Zone | Outside Area |
|---|---|---|---|---|
| Income | In poverty or not | 0.176 | 0.149 | 0.141 |
| Health | Incidence of serious illness and disability | 0.081 | 0.068 | 0.076 |
| | Timely medical treatment | 0.304 | 0.226 | 0.172 |
| Education | Years of education | 0.157 | 0.181 | 0.197 |
| | School attendance | 0.080 | 0.072 | 0.089 |
| Living standard | Drinking water | 0.064 | 0.050 | 0.093 |
| | Fuel | 0.376 | 0.299 | 0.309 |
| | Lavatory | 0.448 | 0.218 | 0.260 |
| | Housing materials | 0.203 | 0.109 | 0.091 |
| | Disposal of garbage | 0.024 | 0.036 | 0.064 |
| Observation | | 125 | 221 | 518 |

2.  Regulation intensity in different geographical locations.

Group inspection is used to further verify whether a significant difference exists in the regulation intensity in different geographical locations (see Table 5). It can be seen from the F value in Table 5 that under different assignment methods, the regulation intensity in the buffer zone is significantly higher than that in the experimental zone and the outside area. The *F* value is used to test the significance of the difference between the mean of two or more samples.

**Table 5.** Group inspection.

| | Buffer Zone | | Experimental Zone | | Outside Area | | F Value |
|---|---|---|---|---|---|---|---|
| | Mean Value | Standard Deviation | Mean Value | Standard Deviation | Mean Value | Standard Deviation | |
| Regulation intensity | 0.538 | 0.172 | 0.509 | 0.196 | 0.446 | 0.198 | 15.71 *** |

Mean values mean the poverty incidence in the buffer zone, experimental zone and outside area, respectively; *** represents the poverty incidence in the different geographical locations is significant different.

In summary, regulation intensity differs in different geographical locations. In a buffer zone, the regulation intensity and the poverty incidence are the highest, with the two

showing correlativity. However, further measurement is necessary to clarify the causal relationship between regulation intensity and the multidimensional poverty of farmers.

### 2.4. Statistics Setting of Model

The explanatory variable, the multidimensional poverty index, which is the focus of this paper, has the lowest limit of 0 and the data have been intercepted. Thus, the tobit model should be used to verify the effect of regulation intensity on the multidimensional poverty of farmers. The specific setting of the model is as follows:

$$y_i^* = \theta + \beta_1 \text{regulation}_i + X_m' \beta_m + \varepsilon_i \varepsilon_i \sim N\left(0, \sigma^2\right) \tag{1}$$

$$y_i = \begin{cases} y_i^*, & \text{if } y_i^* > 0 \\ 0, & \text{if } y_i^* \leq 0 \end{cases} \tag{2}$$

In the formula, $y_i$ is the explanatory variable, that is, the multidimensional poverty index of farmers, $\text{regulation}_i$ is the kernel explanatory variable, i.e., the regulation intensity, $\theta$ is the intercepted item, $\beta_1$ is the solve-for parameter, $\beta_m$ is the vector of the solve-for parameter, $X_m'$ is a group of regulation variables, and $\varepsilon_i$ is the residual term.

### 3. Results

Based on the survey data, the tobit model was used to estimate the effect of regulation intensity on the multidimensional poverty of farmers. The measurement results are shown in Table 6.

**Table 6.** Regulation intensity and the multidimensional poverty of farmers.

| | Multidimensional Poverty Index | |
|---|---|---|
| | **Dimensional Equal Weight** | **Index Equal Weight** |
| Regulation intensity | 0.086 *** | 0.069 *** |
| | (0.027) | (0.024) |
| Age of householder | −0.001 | −0.000 |
| | (0.001) | (0.001) |
| Gender of householder | −0.018 | −0.025 |
| | (0.019) | (0.017) |
| Householder's years of education | −0.012 *** | −0.011 *** |
| | (0.002) | (0.002) |
| Whether householder is a village cadre | −0.007 | −0.006 |
| | (0.015) | (0.013) |
| Family labor scale | −0.004 | −0.003 |
| | (0.005) | (0.004) |
| Upbringing burden of family | 0.099 *** | 0.074 *** |
| | (0.024) | (0.022) |
| Non-farm employment ratio | −0.099 *** | −0.079 *** |
| | (0.020) | (0.018) |
| Farmland | −0.000 | −0.000 |
| | (0.001) | (0.000) |
| Forest land | 0.000 | −0.000 |
| | (0.000) | (0.000) |
| Distance to the town market | 0.006 | 0.005 |
| | (0.004) | (0.004) |
| Region dummy | Yes | Yes |
| Observation | 862 | 855 |

Note: Standard errors in parentheses; *** $p < 0.01$.

### 3.1. Baseline Regression

The regulation intensity will cause a significant increase in the multidimensional poverty index of farmers (Table 6). More precisely, when the dimensional equal weights are used to calculate the multidimensional poverty index of farmers, the multidimensional poverty index will increase by 0.086 for every unit of increase in regulation intensity (Table 6). Moreover, regulation intensity also has a notable positive effect when the index equal weight is used to calculate the multidimensional poverty index of farmers. This finding proved the theoretical analysis described above, that is, the higher the regulation intensity, the greater the restraint upon the rights of the farmers, and thus, the higher the multidimensional poverty index of the farmers. The essence is that the higher the regulation intensity, the more deprivation of the production rights of surrounding farmers, which affected the freedom of production decisions and dampened the production enthusiasm of the surrounding farmers, thereby causing the farmers to reduce the supply of agricultural production and fall into multidimensional poverty.

### 3.2. Robust Test

3.2.1. Using New Dependent Variable

In the preceding part of the paper, only the multidimensional poverty index of farmers was used to measure the state of their poverty. This paper uses "multidimensional poverty or not" to depict the state of farmers' poverty and employs the probit model for verification to avoid the error in the estimation results caused by the measurement. See Table 7 for the estimation results.

**Table 7.** Robust check 1: using new dependent variable.

| | Multidimensional Poverty | |
|---|:---:|:---:|
| | **Dimensional Equal Weight** | **Index Equal Weight** |
| Regulation intensity | 0.103 ** | 0.066 * |
| | (0.041) | (0.035) |
| Age of householder | −0.002 * | −0.001 |
| | (0.001) | (0.001) |
| Gender of householder | 0.023 | 0.051 |
| | (0.036) | (0.034) |
| Householder's years of education | −0.017 *** | −0.019 *** |
| | (0.003) | (0.003) |
| Whether householder is a village cadre | −0.006 | 0.003 |
| | (0.028) | (0.024) |
| Family labor scale | −0.008 | −0.017 ** |
| | (0.008) | (0.007) |
| Upbringing burden of family | 0.104 ** | 0.047 |
| | (0.043) | (0.036) |
| Non-farm employment ratio | −0.068 * | −0.016 |
| | (0.037) | (0.032) |
| Farmland | −0.000 | −0.001 |
| | (0.001) | (0.001) |
| Forest land | −0.000 | −0.000 |
| | (0.000) | (0.000) |
| Distance to the town market | 0.012 | 0.011 * |
| | (0.008) | (0.007) |
| Region dummy | Yes | Yes |
| Observation | 855 | 855 |

Note: Standard errors in parentheses; *** $p < 0.01$, ** $p < 0.05$, * $p < 0.1$.

When dimensional equal weight is used for measurement, the marginal effect coefficient of regulation intensity for the multidimensional poverty of farmers is 0.103, which is significantly positive at the statistical level of 5% (Table 7). Meanwhile, the result is the same when the index equal weight is used for measurement, which shows that with the

increase of regulation intensity, the incidence of farmers' multidimensional poverty will increase significantly, thereby proving the robustness of the estimation result.

### 3.2.2. Using New Independent Variable

The paper further defines the regulation as a binary virtual variable to depict the regulation intensity, with 1 indicating existence and 0 indicating nonexistence. Then, the entropy method was used for a comprehensive evaluation of regulation intensity. See Table 8 for the estimation results.

**Table 8.** Robust check 2: using new independent variable (*k* = 3).

| | Multidimensional Poverty Index | | Multidimensional Poverty | |
|---|---|---|---|---|
| | Dimensional Equal Weight | Index Equal Weight | Dimensional Equal Weight | Index Equal Weight |
| Regulation intensity | 0.046 ** | 0.318 ** | 0.057 ** | 0.053 ** |
| | (0.019) | (0.160) | (0.029) | (0.024) |
| Age of householder | −0.001 | −0.010 * | −0.002 * | −0.001 |
| | (0.001) | (0.005) | (0.001) | (0.001) |
| Gender of householder | −0.017 | 0.139 | 0.025 | 0.052 |
| | (0.019) | (0.202) | (0.036) | (0.035) |
| Householder's years of education | −0.012 *** | −0.090 *** | −0.016 *** | −0.019 *** |
| | (0.002) | (0.019) | (0.003) | (0.003) |
| Whether householder is village cadre | −0.006 | −0.017 | −0.003 | 0.006 |
| | (0.015) | (0.156) | (0.028) | (0.024) |
| Family labor scale | −0.003 | −0.042 | −0.008 | −0.016 ** |
| | (0.005) | (0.045) | (0.008) | (0.007) |
| Upbringing burden of family | 0.099 *** | 0.571 ** | 0.103 ** | 0.047 |
| | (0.024) | (0.237) | (0.043) | (0.035) |
| Non-farm employment ratio | −0.100 *** | −0.387 * | −0.070 * | −0.017 |
| | (0.020) | (0.207) | (0.037) | (0.032) |
| Farmland | −0.000 | −0.002 | −0.000 | −0.001 |
| | (0.001) | (0.005) | (0.001) | (0.001) |
| Forest land | 0.000 | −0.000 | −0.000 | −0.000 |
| | (0.000) | (0.000) | (0.000) | (0.000) |
| Distance to the town market | 0.006 | 0.069 | 0.012 | 0.011 * |
| | (0.004) | (0.042) | (0.008) | (0.006) |
| Region dummy | Yes | Yes | Yes | Yes |
| Observation | 862 | 855 | 855 | 855 |

Note: Standard errors in parentheses; *** $p < 0.01$, ** $p < 0.05$, * $p < 0.1$.

The estimation results (Table 8) from the re-measurement of regulation intensity are consistent with those of the baseline regression (Table 6), further proving the robustness of the estimation results.

### 3.3. Heterogeneity Analysis

The type of regulation of nature reserves can be divided into the regulation of agricultural production and the regulation of pollutants. The former relates directly to the farmers' production. The higher the regulation intensity of agricultural production, the greater the restraint upon farmers and the more likely they will fall into multidimensional poverty. Comparatively, the latter will produce an indirect effect upon the multidimensional poverty of farmers. This paper further divides the regulation into the regulation of agricultural production and the regulation of pollutants to verify whether any heterogeneity exists in the effect of different types of regulation on the multidimensional poverty of farmers. See Table 9 for the estimation results.

**Table 9.** Different types of regulation and the multidimensional poverty of farmers ($k = 3$).

| | Multidimensional Poverty Index | | | | Multidimensional Poverty | | | |
|---|---|---|---|---|---|---|---|---|
| | Dimensional Equal Weight | | Index Equal Weight | | Dimensional Equal Weight | | Index Equal Weight | |
| Regulation of agricultural production | 0.101 *** (0.032) | | 0.081 *** (0.028) | | 0.677 ** (0.273) | | 0.580 * (0.306) | |
| Regulation of pollutants | | 0.548 *** (0.172) | | 0.440 *** (0.154) | | 3.666 ** (1.475) | | 3.138 * (1.656) |
| Control variables | Yes | Yes | Yes | Yes | Yes | Yes | Yes | Yes |
| Region dummy | Yes | Yes | Yes | Yes | Yes | Yes | Yes | Yes |
| Observation | 855 | 855 | 855 | 855 | 855 | 855 | 855 | 855 |

Note: Standard errors in parentheses; *** $p < 0.01$, ** $p < 0.05$, * $p < 0.1$.

Table 9 shows that the regulation of agricultural production and the regulation of pollutants produce a significantly positive effect on the multidimensional poverty of farmers. The influence coefficient of the latter is higher than the effect of the former. The estimation results are also consistent with whether they were obtained using the multidimensional poverty index, the question of whether multidimensional poverty exists, and the method of different weights, thereby further proving the robustness of the estimation results.

*3.4. Analysis of Mechanism*

The theoretical analysis indicates that the regulation has directly deprived the surrounding farmers of their rights, thereby impairing their freedom of production decisions. It not only causes the farmers to reduce the supply of agricultural production but also damages the inherent incentive mechanism for the surrounding farmers, which results in the lack of the inducement to utilize resources for optimized allocation and fall into the vicious spiral of poverty. Thus, this paper has incorporated farmers' freedom of production decisions into the model to verify the logic "regulation intensity–freedom of production decision–multidimensional poverty". Specifically, a five-level scale was used to measure the freedom of the production decision, with 1 indicating high dissatisfaction and 5 indicating high satisfaction. See Tables 10 and 11 for estimation results.

**Table 10.** Regulation in nature reserves vs. multidimensional poverty index of farmers.

| | Multidimensional Poverty Index [1] | Freedom of Production Decision | Multidimensional Poverty Index [1] | Multidimensional Poverty Index [2] | Freedom of Production Decision | Multidimensional Poverty Index [2] |
|---|---|---|---|---|---|---|
| Regulation intensity | 0.088 *** (0.030) | −0.648 *** (0.159) | 0.070 ** (0.030) | 0.084 ** (0.027) | −0.648 *** (.159) | 0.073 *** (0.028) |
| Freedom of production decision | | | −0.027 *** (0.009) | | | −0.016 ** (0.008) |
| Control variables | Yes | Yes | Yes | Yes | Yes | Yes |
| Region dummy | Yes | Yes | Yes | Yes | Yes | Yes |
| Observation | 486 | 486 | 486 | 486 | 486 | 486 |
| Indirect effect | | 0.018 | | | 0.010 | |
| Direct effect | | 0.070 | | | 0.073 | |
| Total effect | | 0.088 | | | 0.084 | |

Note: Standard errors in parentheses; *** $p < 0.01$, ** $p < 0.05$; [1] dimensional equal weight; [2] index equal weight.

**Table 11.** Regulation in nature reserves vs. multidimensional poverty index of farmers (using new main independent variable).

| | Multidimensional Poverty Index [1] | Freedom of Production Decision | Multidimensional Poverty Index [1] | Multidimensional Poverty Index [2] | Freedom of Production Decision | Multidimensional Poverty Index [2] |
|---|---|---|---|---|---|---|
| Regulation intensity | 0.064 *** (0.023) | −0.388 *** (0.119) | 0.053 ** (0.023) | 0.063 *** (0.020) | −0.388 *** (0.120) | 0.057 *** (0.020) |
| Freedom of production decision | | | −0.028 *** (0.009) | | | −0.016 ** (0.008) |
| Control variables | Yes | Yes | Yes | Yes | Yes | Yes |
| Region dummy | Yes | Yes | Yes | Yes | Yes | Yes |
| Observation | 486 | 486 | 486 | 486 | 486 | 486 |
| Indirect effect | | 0.011 | | | 0.010 | |
| Direct effect | | 0.053 | | | 0.057 | |
| Total effect | | 0.064 | | | 0.063 | |

Note: Standard errors in parentheses; *** $p < 0.01$, ** $p < 0.05$; [1] dimensional equal weight; [2] index equal weight.

Table 10 shows the regulation intensity results measured with the weight assignment method 1. The higher the regulation intensity, the higher the restraint upon farmers' freedom of production decisions and the easier they fall into multidimensional poverty. The logical mechanism described above has been verified and fully proved that the regulation in nature reserves will have a direct effect on farmers' freedom of production decisions, making it difficult for them to become rich.

Table 11 shows the regulation intensity results measured with weight assignment method 2, which are the same as those of Table 10, thereby proving the robustness of the estimation results.

## 4. Discussion

### 4.1. Multidimensional Poverty

Poverty is a dynamic and historical comprehensive concept. With the evolution of social and economic development, the connotation of poverty is constantly changing, and it has experienced the evolution from single poverty to multidimensional poverty [47]. Specifically, the early recognition of poverty in academia is often limited to the lack of subsistence materials, and usually only focuses on the survival needs of individuals, which is obviously absolute and objective. In addition, the traditional poverty theory treats income as a pure economic phenomenon, ignoring the essence behind income inequality. However, with the progress of humankind and the development of social economy, the early understanding of poverty is difficult to adapt to the changes of social and economic development.

In the 1980s and 1990s, Amartya Sen proposed the "feasible ability" theory, starting from the economic and social dimensions, and establishing a new analytical framework for studying poverty. Sen believes that insufficient income is only the resultant manifestation of the poverty of individual feasible ability, and lack of ability is the true connotation and essential characteristic of the poverty of individual feasible ability [48]. Therefore, poverty not only refers to the consumption or income level, but also includes multiple dimensions such as education, health, and living standards [49].

Additionally, with the solution of poverty and the deepening of understanding, the political and academic circles have begun to take the issue of multidimensional poverty seriously [50]. Our article can provide a policy reference for how the government can solve the multidimensional poverty of farmers by implementing reasonable regulation intensity, especially how to strengthen regulation while avoiding restrictions on farmers' freedom of production decisions, ensuring flexible resource allocation, and alleviating poverty. This paper combines existing research to construct a multidimensional poverty index with four dimensions of income, health, education, and living standards, in order to comprehensively analyze the current poverty situation of farmers around PNRs in China.

*4.2. Regulation Intensity and Multidimensional Poverty*

The key to eliminating poverty and achieving common prosperity is to achieve a harmonious coexistence between human and nature. Existing studies generally recognize the importance of establishing nature reserves and have focused on the relationship between nature reserves and multidimensional poverty. However, the existing literature on regulation areas and poverty has not reached a consensus.

As ecological environment protection restricts the development and utilization of natural resources, most studies point out that the establishment of regulation areas has aggravated the poverty of local farmers. Research from the perspective of cost–benefit analysis generally believes that farmers in surrounding communities bear the huge economic and social costs of protecting natural resources, but the corresponding benefits from regulation areas have not increased significantly. The establishment of regulation areas reduces the economic welfare of local residents [37]. In addition, most studies focusing on the livelihoods of farmers also believe that the establishment of regulation areas has brought significant negative impacts on farmers' livelihoods (such as resource constraints, conflicts between humans and wild animals, migration and relocation, etc.), which has aggravated the poverty of farmers [51,52]. However, other studies have pointed out that regulation areas also have a positive impact on farmers' livelihoods, such as direct ecological compensation, non-agricultural employment, eco-tourism, and increased infrastructure investment [24]. Therefore, the establishment of regulation areas reduced the incidence of poverty due to sufficient ecological and environmental benefits.

In addition, most studies analyze from a dichotomous angle of "whether" regulations exist and pay less attention tothe effects of poverty caused by the difference in the regulation intensity of the same region [18]. However, in practice, the regulation system arrangements and implementation of different protected areas are heterogeneous, resulting in significant differences in the impact of protected areas on the poverty of farmers. This paper focuses on the regulation intensity to verify the relationship between regulation and multidimensional poverty and provides a logically consistent explanation for the divergence of existing research. There is ample evidence, from our regression results, that the higher the regulation intensity, the easier it is for farmers to fall into multidimensional poverty. The mechanism is that the regulation system restricts farmers' freedom of production decisions, which makes it impossible to achieve optimal allocation of resources, leading to poverty.

Additionally, with the solution of poverty and the deepening of understanding, the political and academic circles have begun to take the issue of multidimensional poverty seriously [50]. Our article can provide a policy reference for how the government can solve the multidimensional poverty of farmers by implementing reasonable regulation intensity, and especially how to strengthen regulation while avoiding restrictions on farmers' freedom of production decisions, ensuring flexible resource allocation, and alleviating poverty.

However, because of the limitations of data, this paper uses panel data only for empirical analysis and is therefore unable to observe the dynamic effects of regulation on the multidimensional poverty of farmers, which can be supplemented by future studies.

## 5. Conclusions

This paper uses the farmers around PNRs in China as the objects of study to explore the effects of the regulation intensity of the reserve upon the multidimensional poverty of surrounding farmers and its inherent mechanism from the two levels of theory and empirical evidence. Our study found that regulation intensity has significantly increased the multidimensional poverty of the farmers. Specifically, when the multidimensional poverty index of farmers is measured with the weights of dimensions, the regulation intensity increases by one unit, and the multidimensional poverty index significantly increases by 0.086 at the statistical level of 1%; when the multidimensional poverty index of farmers is calculated with the weights of the indicators, the regulation intensity increases every time by one unit, and the multidimensional poverty index significantly increases by 0.069 at the statistical level of 1%. Moreover, the regulation of agricultural production and the regu-

lation of pollutants will produce a significantly positive effect on their multidimensional poverty, and the inherent mechanism of action is that regulation intensity has restricted farmers' freedom of production decisions, thereby causing them to sink into poverty. In other words, the greater the intensity of regulation, the more restricted the freedom of farmers' production decisions, and the easier it is to fall into multidimensional poverty. The conclusion of this paper has not only expanded the theoretical research on regulation and multidimensional poverty but also provided policy suggestions to boost the socialist modernization of China and achieve common prosperity.

First, the government should see that regulations have pros and cons. On the one hand, the regulation of rights in nature reserves can help the ecological environment to recover and protect animal diversity; on the other hand, such regulations may also affect the freedom of production decisions of surrounding farmers, thereby causing them to fall into multidimensional poverty. Policies should properly control the intensity of regulations to avoid over-regulation and zero regulation. Monitoring needs to be implemented in light of the local reality in a year and receive a dynamic adjustment.

Second, the government should first understand that poverty has multiple dimensions so that farmers' poverty cannot be measured based merely on their income or state of livelihood. For the farmers around a nature reserve, the restraint upon their right to freedom of production decisions is the reason why they fall into multidimensional poverty. Therefore, the government should scientifically formulate the content of regulations and further clarify the rights of the farmers based on protecting nature reserves rather than adopt a sweeping approach.

Lastly, the government should implement a matching compensation policy while regulating the farmers around nature reserves, provide capable farmers with employment training and guidance to help them realize the optimized allocation of resources, including labor force and land, and form the inherent drive to remove poverty, and provide disadvantaged farmers with more support and compensation in relocation and medical care, among others, and use government support to help them break away from poverty.

**Author Contributions:** Conceptualization, C.L. and L.G.; Methodology, C.L. and L.G.; Software, C.L.; Validation, C.L. and L.G.; Formal analysis, C.L. and L.G.; Investigation, C.L. and L.G.; Data collection, C.L. and L.G.; writing—original draft preparation, C.L.; writing—review and editing, C.L. and L.G.; supervision, C.L. and L.G.; project administration, L.G.; funding acquisition, L.G. All authors have read and agreed to the published version of the manuscript.

**Funding:** This research was supported by the National Natural Science Foundation of China and the Consultative Group for International Agricultural Research "The impact of habitat regulatory policies on ecological protections and rural livelihoods: The case of giant panda protected areas in China", grant number 7171101101.

**Institutional Review Board Statement:** Not applicable.

**Informed Consent Statement:** Not applicable.

**Data Availability Statement:** The data could provide by the authors if necessary.

**Acknowledgments:** We greatly acknowledge Wei Duan and Wei Zhou for their help in setting up the experiment.

**Conflicts of Interest:** The authors declare no conflict of interest. The funders had no role in the design of the study; in the collection, analyses, or interpretation of data; in the writing of the manuscript, or in the decision to publish the results.

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
