# Peer review of "Regulation Intensity, Freedom of Production Decision and the Poverty of Farmers: Evidence from the Panda Nature Reserves in China"

_forests, doi:10.3390/f12111528_

Round 1
Reviewer 1 Report
The article is valuable in both theoretical and practical terms. I appreciate it perfectly.
In the future, perhaps I would recommend describing the results below the tables a little more.
Author Response
Dear Reiewer,
Thanks for your valuable suggestions. As suggested, we have added the describing contents of tables in this paper. Please see the second paragraph part on Page 6 and first paragraph part on Page 8.
Many thanks.
Best Wishes.
Reviewer 2 Report
The paper is intersting, but it needs to be imrpoved further.
Main remarks:
Introduction
L77-107-Consider adding more references at international level
"2 Source of Data"-More information may be added about the structe ofthe questionaire.
"Hence, this paper uses the entropy method to calculate the regulation intensity based on"-There are other alternatives of aggregation. You may mention them. The problem of aggregation is well discussed among several autors and the are alternantive ways.
2.3.3. Control variables-OK. Well selected.
The rresults are well presented and are scientifically sound.
Author Response
Dear Rewiewer,
Thank you for your effort on our manuscript.
The attached document is our responses to your advises and suggestions.
Many thanks.
Best Wishes.

Reviewer 3 Report
Very interesting. Only scertain points of improvement:
1) check the grammar/syntax again (e.g. "when they 134
use of crucial resources is placed under control...")
2) methodologically, clarify where you have found the list of the whole population in order to carry out a random sampling
3) the indicators K, H, A, M (lines 214-216) should be explained
4) how the equal weights weree measured and what they practically mean should be also explained.
5) put highlights in the conclusion/ point out the main findings and present them in a way so as to make them non-trivial.
Author Response
Dear Reviewer,
Thank you for your time on our paper, and giving us the valuable advises.
The attached document is our responses to your opinion.
Thank you very much.
Best Wishes.
